# Dissecting heritability, environmental risk, and air pollution causal effects using > 50 million individuals in MarketScan

Daniel McGuire[1,7], Havell Markus[2,3,4,7], Lina Yang[1], Jingyu Xu[1], Austin Montgomery[2], Arthur Berg[1], Qunhua Li [5], Laura Carrel [6], Dajiang J. Liu [1] ✉ & Bibo Jiang [1] ✉

Large national-level electronic health record (EHR) datasets offer new opportunities for disentangling the role of genes and environment through deep phenotype information and approximate pedigree structures. Here we use the approximate geographical locations of patients as a proxy for spatially correlated community-level environmental risk factors. We develop a spatial mixed linear effect (SMILE) model that incorporates both genetics and environmental contribution. We extract EHR and geographical locations from 257,620 nuclear families and compile 1083 disease outcome measurements from the MarketScan dataset. We augment the EHR with publicly available environmental data, including levels of particulate matter 2.5 ($PM_{2.5}$), nitrogen dioxide ($NO_2$), climate, and sociodemographic data. We refine the estimates of genetic heritability and quantify community-level environmental contributions. We also use wind speed and direction as instrumental variables to assess the causal effects of air pollution. In total, we find $PM_{2.5}$ or $NO_2$ have statistically significant causal effects on 135 diseases, including respiratory, musculoskeletal, digestive, metabolic, and sleep disorders, where $PM_{2.5}$ and $NO_2$ tend to affect biologically distinct disease categories. These analyses showcase several robust strategies for jointly modeling genetic and environmental effects on disease risk using large EHR datasets and will benefit upcoming biobank studies in the era of precision medicine.

It is widely known that most complex traits are jointly influenced by genetics and environment. Yet, the extent to which genetic or environmental factors contribute to complex traits is much less understood and often subject to contentious debate[1], possibly due to the lack of large and high-quality datasets containing both genetic data and environmental exposures. Quantifying the genetic and environmental contributions to human disease is critical to understanding the underlying biology, performing accurate risk predictions, and designing effective preventive and therapeutic interventions.

Traditionally, family-based studies and variance components models have been used to partition phenotypic variance into genetic and environmental components, where health outcome similarities among relatives are regressed over measures of genetic relatedness and shared environmental exposure[2,3]. In these studies, unmeasured

[1]Department of Public Health Sciences, Penn State College of Medicine, Hershey, PA 17033, USA. [2]MD/PhD Program, Penn State College of Medicine of Medicine, Hershey, PA 17033, USA. [3]Bioinformatics and Genomics PhD Program, Penn State College of Medicine, Hershey, PA 17033, USA. [4]Institute for Personalized Medicine, Penn State College of Medicine, Hershey, PA 17033, USA. [5]Department of Statistics, Penn State University, University Park, PA, USA. [6]Department of Biochemistry and Molecular Biology, Penn State College of Medicine, Hershey, PA 17033, USA. [7]These authors contributed equally: Daniel McGuire, Havell Markus. ✉e-mail: dajiang.liu@psu.edu; bjiang@phs.psu.edu

**Table 1 | Demographic description of ascertained nuclear families in the Marketscan database**

| | |
|---|---|
| **Total # of families** | 257,620 |
| **Unique family locations (County or MSA)** | 3,229 |
| **Median number of families per location (IQR)** | 15 (5,50) |
| **Median months enrolled (IQR)** | 84 (72,102) |
| **Median age at first year of enrollment (IQR)** | |
| Overall | 23.5 (15,46) |
| Father | 47 (44,51) |
| Mother | 45 (42,49) |
| Children | 15 (13,17) |
| **Median age difference (IQR)** | |
| Father – Mother | 1.907 (0,4) |
| Oldest Sibling – Youngest Sibling | 3.061 (2,4) |
| **% Primary enrollee by parent** | |
| Father | 67.3% |
| Mother | 32.7% |
| **Fraction of smokers** | |
| Overall | 66,723 (6.5%) |
| Father | 23,457 (9.1%) |
| Mother | 17,643 (6.8%) |
| Children | 25,623 (5%) |

*IQR* inter-quartile range

environmental exposures across different families are often assumed to be independent. However, a number of environmental risk factors are shared between different families in the same geographic area and are spatially correlated[4]. Examples include air pollution, climate, and sociodemographic characteristics such as average levels of education and income[5,6]. Unmodeled community-level environment effects could lead to upwardly biased estimates of genetic heritability, as within-family phenotypic correlation due to shared community-level environment may be falsely attributed to genetics[4]. Twin studies may also be subject to the impact of environmental confounding, and often do not estimate the contribution of community-level environment.

Genome-wide association studies (GWAS) and linear mixed models with unrelated individuals have been used to estimate narrow-sense heritability that is captured by the genotyped SNPs (chip heritability)[7]. GWAS using unrelated individuals often achieve much larger sample sizes and are less likely to be confounded by shared environment compared to family-based studies[8]. However, recent research has shown that geographical confounding can bias estimates of chip heritability as well[9]. Besides, these chip heritability estimates are conceptually different from that of family-based studies and are sensitive to the assumptions on the allele frequencies, effect sizes, and levels of linkage disequilibrium between genotyped SNPs and the true causal variants[8]. Standard linear mixed models in GWAS often do not model community-level shared environmental variance and hence do not quantify its contributions to disease. There are also existing works that seek to estimate environmental impacts on diseases but do not account for genetic relatedness, and hence do not provide joint estimates of heritability and environmental contribution either[3,10–15].

In this work, to address the aforementioned challenges and fill in the knowledge gap, we describe a spatial mixed linear effect (SMILE) model that jointly estimates genetic heritability and environmental components of disease risk using geospatial locations of the study participants as a proxy for community-level environmental risk factors. The SMILE model helps characterize geographic variation in disease risk, control for additional sources of within-family correlation, and reduce the bias of estimated heritability. We apply SMILE to the MarketScan dataset[16], a large insurance billing database with electronic

health records (EHR) from more than 50 million individuals to assess the contribution of genetic and environmental factors to 1083 human diseases. To further assess if environmental risk factors are causally linked to disease, we augmented the MarketScan dataset with publicly available environmental data, including levels of particulate matter 2.5 ($PM_{2.5}$) and nitrogen dioxide ($NO_2$), climate, and socioeconomic status. We apply a rigorous causal inference framework to assess the roles of pollutants $PM_{2.5}$ and $NO_2$ for the phenome using wind speed and direction as instrumental variables.

## Results data overview
We used the IBM MarketScan health insurance claims database to assemble a large quality-filtered cohort of 257,620 nuclear families with parents and children (Methods). The MarketScan database is a de-identified compilation of patient billing code records from employer-based health insurance policies in the United States. Family structure was inferred based on the relationship of each member to the primary enrollee on the policy. Members indicated as either "Employee" or "Spouse" were deemed as parents and those indicated as "Child/Other" were deemed as children in the family[17]. We analyze nuclear families who were enrolled in the database for at least 6 years between 2005 and 2017, and for whom all children are at least 10 years old at the time of entry into the database. As we demonstrate in the Results and Supplementary Methods, the impact of mis-specified familial relationship and the length of enrollment in the database have minimal impact on the estimates of variance components.

A summary of the available demographic characteristics of ascertained families is provided in Table 1. We then mapped the inpatient and outpatient (International Classification of Diseases version 9 and 10 (ICD-9/ICD-10) billing code records between 2005 to 2017 to PheWAS codes[13], which represent biologically/medically more meaningful phenotypes. We also include several individual-level covariates including the year of birth and sex, as well as the approximate location in terms of U.S. County or Metropolitan Statistical Area (MSA) in our analyses.

Multiple external datasets of community-level risk factors were also incorporated, which were assigned to each individual based on their location. These include demographic data extracted from the 2015 American Community Survey 5-year estimates[18], satellite-derived measurements of air pollution including particulate matter 2.5 ($PM_{2.5}$)[19,20] (Supplementary Fig. 1) and nitrogen dioxide ($NO_2$) (Supplementary Fig. 2-3)[21,22]. We also integrated wind direction and wind speed data[23] as instrumental variables for causal inference (see Methods for details).

### Statistical methodology overview
In the SMILE model, we incorporate random effects to capture phenotypic variation attributable to genetic relatedness, shared family environment, and shared community-level environment. The full SMILE model is specified by

$$Y = X\pi + u_g + Z_s u_s + Z_{par} u_{par} + Z_{child} u_{child} + \epsilon$$

In this model:
- $Y$ is the vector of 0-1's for disease status, with 1 being the diseased and 0 being the normal.
- $X$ is the design matrix for fixed effect individual-level covariates with effects $\pi$.
- $u_g$ is the vector of genetic random effects, whose correlation is determined by genetic relatedness. Individuals in different families are assumed to be genetically unrelated.
- $u_{par}$ and $u_{child}$ are the vectors of random effects for the shared parental and children-level family environment. Even though they live in the same household, parents and children may share distinct environment, including diet patterns, exercise levels,

hours of sleep, and exposures at work or school. $u_{par}$ and $u_{child}$ each capture the distinct environmental exposures that are shared between parents and between children.

- $Z_{par}$ and $Z_{child}$ are the design matrices that link each individual to their corresponding within-family parent- or child-shared environment random effect.
- $u_s$ is the random effect for the community-level shared environment. Families from the same location share the same random effect. We assume that the random effects from neighboring locations follow an independent normal distribution (IND), or conditional autoregressive (CAR), or simultaneous autoregressive (SAR) distribution. We chose these model specifications as they cover a wide range of scenarios and are computationally feasible for large-scale datasets.
- $Z_s$ is the design matrix linking each individual to his/her corresponding spatial random effects.

We also extended the model in a two-stage regression framework (SMILE-2) to assess the causal effects of air pollution on the phenome, which is conceptualized in Supplementary Fig. 4. We use wind speed and direction as instrumental variables[24]. Wind speed and direction have previously been used as instrumental variables for various pollution exposures[25–28]. They are unlikely to have any direct causal effect on a disease phenotype, but are strong predictors of local pollution levels, as the pollution level in a given location is a mixture of locally produced and transported air pollution by the wind from its original source[28,29].

It is easy to verify that the wind speed and direction satisfy the three primary assumptions of instrumental variables in two-stage regression analyses[24]:

- Instrument relevance: the instrument is correlated with the pollutant $P$, i.e., $\text{cov}(Z,P) \neq 0$, where $Z$ collectively denotes for wind speed and direction.
- Instrument exogeneity: The instrument (averaged wind-direction) is uncorrelated with other confounders (measured or unmeasured) in the second stage model.
- The averaged wind instruments $Z$ have no direct effect on the disease phenotype $Y$.

In the first stage regression, we regress the pollution levels over the instrumental variables. In the second stage model, the SMILE-2 model tests for the causal effect of pollution ($\beta$) using the predicted pollution level ($\widetilde{P}$) from the first stage model as input, i.e.,

$$Y = \widetilde{P}\beta + X\pi_2 + u_g + Z_{par}u_{par} + Z_{child}u_{child} + \epsilon$$

More details on SMILE and SMILE-2 models can also be found in Methods and the Supplementary Methods.

## SMILE model yields more accurate heritability and environmental variance component estimates

We conducted extensive simulations to assess the accuracy of the variance component estimates when models are correctly- or mis-specified (Methods). To make sure that our simulation reproduces realistic spatial distributions of family locations, disease prevalence, risk factors, and confounders, we sampled nuclear families with replacement from the available locations in the MarketScan dataset based on the 257,620 families used in data analysis. We varied the values of different variance components in the simulation and used CAR covariance structure for simulating spatial random effects as it best fits the data (shown in the sections below). For each combination of variance components, we simulated the underlying liability and created binary phenotypes under the liability threshold model with varying disease prevalence. We compared sub-models with different combinations of random effects in the simulation. A total of eight

models were fitted in the analysis, including *SMILE (GPC + S), GP + S, GC + S, GPC, PC + S, PC, G + S,* and *S,* where we use *G, P, C,* and *S* to denote the genetic, parental, children's, or spatial community-level variance components. We used Bayesian Information Criterion (BIC) to determine the best fitting model.

We found that BIC chose the model with correctly specified variance components in 71.1% of the replicates with 50,000 quad-families, and 81.8% of replicates with 250,000 quad-families (Fig. 1A). The estimates became more precise as sample size and prevalence increase (Supplementary Data 1). We found that in the presence of community-level spatial effects, the models that do not account for community-level effects produce upwardly biased heritability estimates, as family members who live in the same location have additional phenotypic correlation which may be falsely attributed to genetics (Fig. 1B, Supplementary Data 1). Our results show that the extent of upward bias in heritability increases with the size of the spatial variance components. Importantly, the full SMILE model yielded minimal or near-minimal bias and mean squared error (MSE) for all variance component estimates regardless of the underlying model (Fig. 1C, Supplementary Data 1). For instance, if the true model does not contain parental shared family environment, SMILE still gave unbiased estimates for all variance components. For this reason, we used the full model with *GPC + S* variance components in our analysis of MarketScan data, as it eases the computational burden of estimating multiple models without compromising the estimation accuracy. We also verified that our chosen parameters for simulations reflect the parameters estimated from the MarketScan and that our simulation setting is realistic (Supplementary Data 2). We lastly conducted simulations with family relationship errors (e.g., when stepchildren and adopted children are coded as biological children for both parents) to assess the robustness of heritability estimates (see Supplementary Methods for more details). Overall, we observed that the heritability estimates from SMILE models are robust. With noisy familial relationships, the confidence intervals of the estimated variance components still overlap the true values and the bias is small (Supplementary Fig. 5 and Supplementary Data 3).

## SMILE-2 model more powerfully identifies pollution causal effects

Similar to variance component simulation, we sampled with replacement families and their covariates from the MarketScan dataset, so that all confounders and the covariates of interest (i.e., pollution levels) maintain their realistic correlations. To assess the type-I error and power for causal inference, we simulated the pollution effects of either $NO_2$ or $PM_{2.5}$, varying the relative risk (RR). Based on estimated variance components and assumed causal effects, we simulated disease liabilities, which were dichotomized to get binary disease status. For each PheWAS code, we simulated six replicates and ensured that each PheWAS code-based phenotype was selected under both the null (RR = 1) and alternative (RR > 1) hypotheses at least once.

To illustrate the power for SMILE-2, we also compared the results of SMILE-2 with a standard two-stage regression model of independent individuals with fixed effects only (*IND-FE*).

Both *IND-FE* and *SMILE-2* Models produced calibrated Type-I error rates under the null hypothesis for each significance threshold (Fig. 2A). The *SMILE-2* model was substantially more powerful than *IND-FE* models, particularly when disease prevalence is low, or when the disease prevalence is moderate and the causal effect (measured by RR) is large (Fig. 2A), as *SMILE-2* incorporates additional related samples. MSEs of the estimated log-odds ratios are also lower for *SMILE-2* compared to *IND-FE* (Fig. 2B). These results indicate that modeling genetically related individuals in insurance claim data using the *SMILE-2* model enlarges sample size and improves the accuracy of causal effect estimates and the power of causal inference. We also conducted simulations to assess how the causal effects of pollution may be

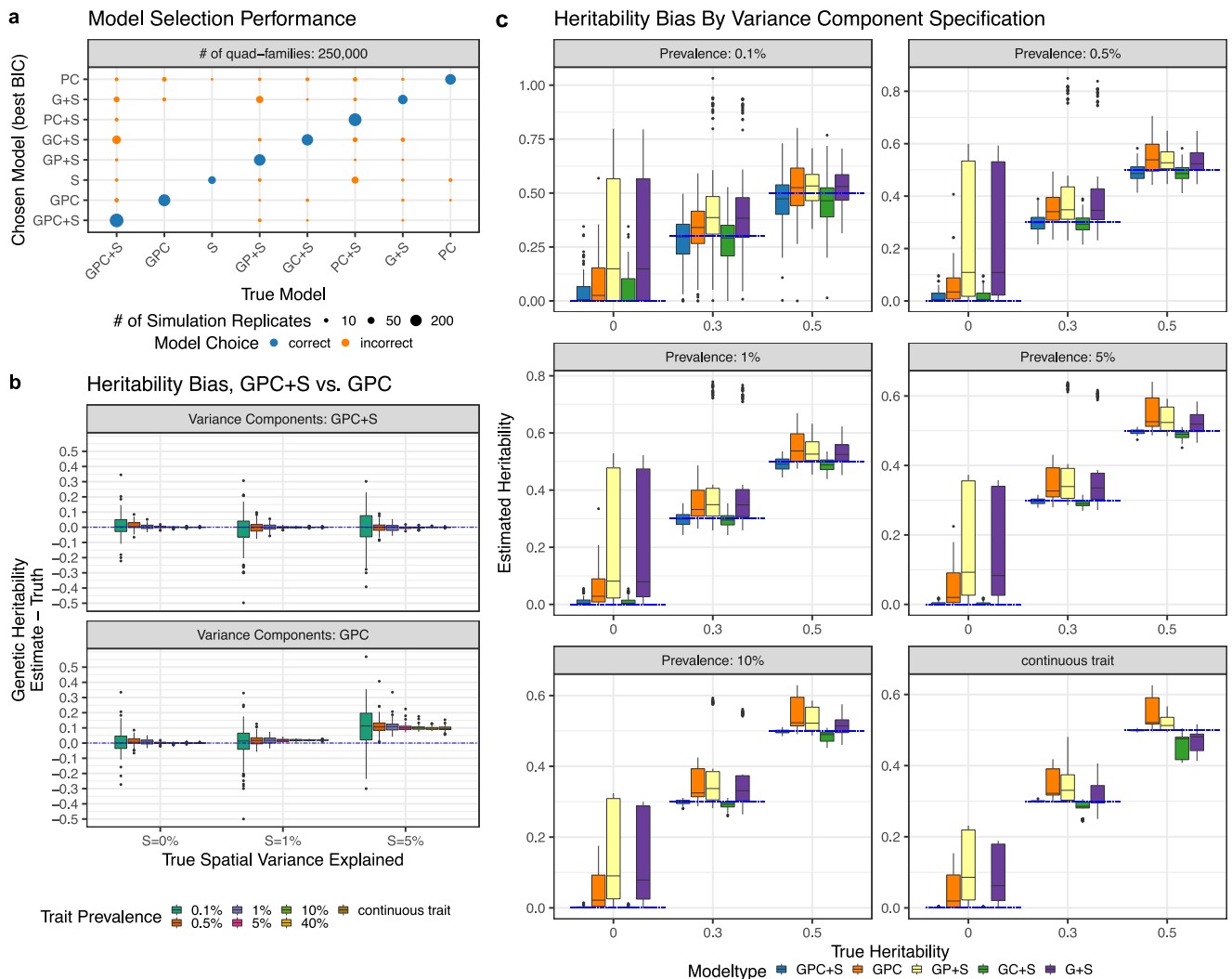

**Fig. 1 | Variance component model simulations.** We conducted comprehensive simulations to evaluate the SMILE model for estimating heritability and environmental variance components under different model specifications. We simulated data under models with different combinations of genetic (G), parental (P), children (C) and spatial community-level (S) environment variance components. When analyzing simulated data, Bayesian information criterion (BIC) was used to determine the best model. In **a**, we display the fraction of replicates where each model was selected, as reflected by the size of the data point. In **b**, we compare GPC and GPC + S models as the amount of true community-level spatial variance increases. Each dot represents the difference in estimated genetic heritability and true simulated heritability across 5 simulation replicates under 60 different scenarios. In **c** we showed the accuracy for the heritability estimates from SMILE under different true models with varying numbers of variance components. Each dot represents the estimated genetic heritability across 5 simulation replicates under 60 different scenarios. In **b** and **c**, minima and maxima values (excluding outliers) are represented by the lower- and upper-bound of the whiskers. Median value is represented by the bold line in the middle. First and third quartiles are represented by the lower- and upper-bound of the box. All panels represent simulations using 250,000 nuclear families.

---

affected if the pollution and wind measurements are noisy (see Supplementary Methods for more details). We observed that causal effect estimates by SMILE-2 remain unbiased when noisy pollution and wind measurements were used (Supplementary Fig. 6) and the power is only minimally reduced.

### Estimation and robustness of genetic heritability and spatial variance components for 1083 traits

We used the SMILE model to analyze 1083 binary diseases as defined by the PheWAS code (Supplementary Data 2). Comparing the GPC and SMILE models, we found that BIC values were generally smaller for the full SMILE model (with spatial random effects) in 1021/1,083 (94.3%) of phenotypes, which suggests that modeling community-level environment improves model fitting. We compared estimated genetic heritability from the best models chosen for each phenotype with or without spatial random effects (Fig. 3A). When the spatial variance component was added to the model, the estimated heritability

generally decreases, with the median decrease being 0.03 and interquartile range (IQR) being (0.018, 0.051). This verifies that many complex traits are influenced by the shared community-level environment, and that the failure to model the shared community-level environment could lead to inflated heritability estimates.

Among the 1,021 traits for which the full SMILE model was chosen as the best model, a CAR covariance structure was selected for 783 (76.7%) traits, compared to 203 (19.9%) for SAR and 35 (3.4%) for uncorrelated covariance structures. We compared the heritability and spatial variance component estimates for SAR and IND against the CAR models for each phenotype (Fig. 3B, C). Interestingly, the estimated heritability was virtually identical regardless of the correlation structure of the spatial random effect (mean absolute difference compared to CAR = 0.002, with standard deviation 0.003). Differences in the spatial variance component estimates were also small (mean absolute difference compared to CAR = 0.002 with standard deviation 0.002). This is an indication that

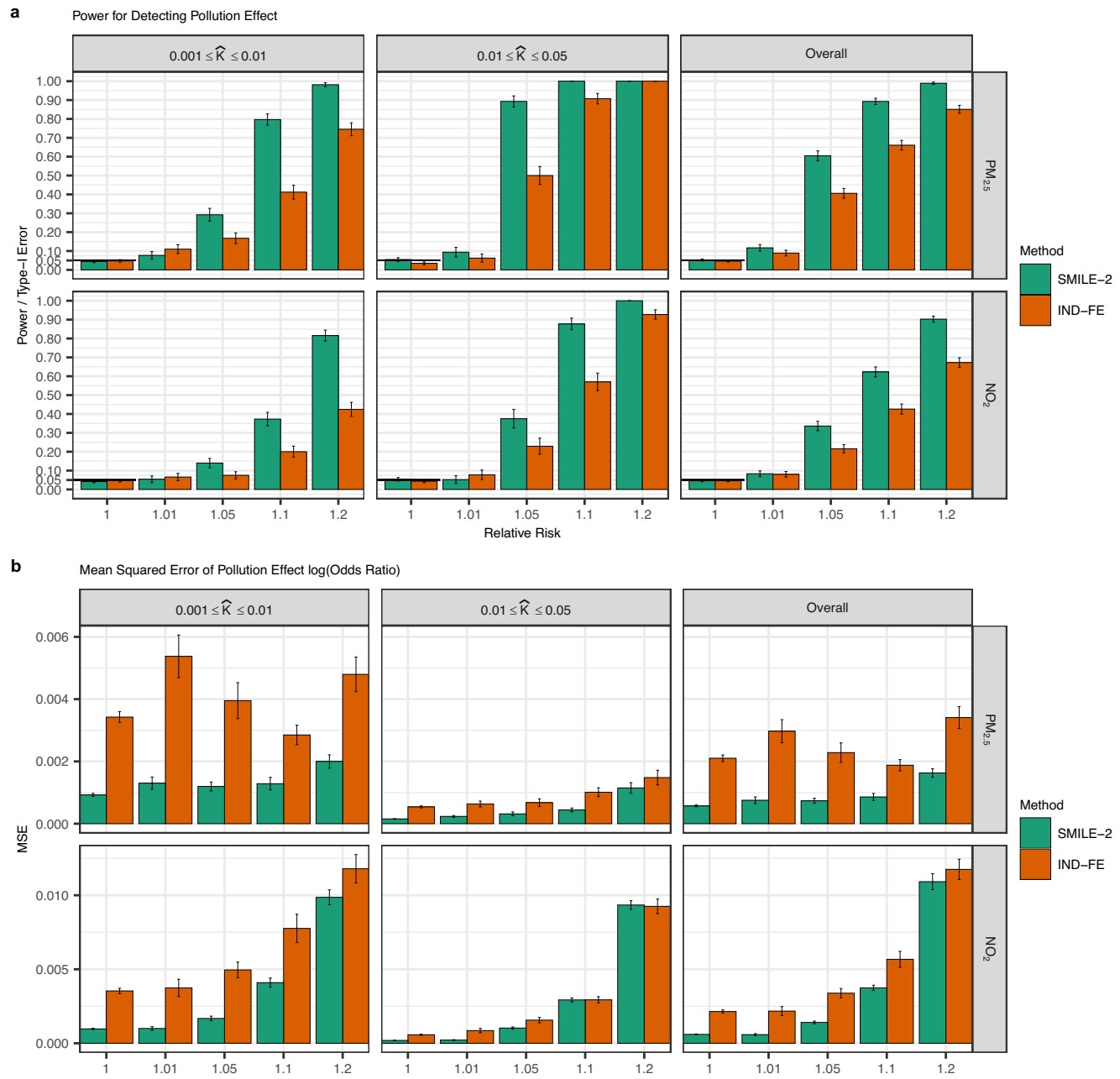

**Fig. 2 | SMILE-2 controls type I error and improves power over methods that ignore genetic relatedness.** We simulated scenarios with 250,000 quad families. We resampled with replacement families from MarketScan as well as the confounder variables associated with those families. We varied the causal effect (in the unit of relative risk) for PM$_{2.5}$ and NO$_2$ between 1.0, 1.01, 1.05, 1.1, or 1.2. Confounding effects as well as genetic and family environment variance components were simulated based on the parameter estimates reflective of reported values for complex traits. The binary disease status was obtained by dichotomizing the continuous liability threshold according to the disease prevalence from real data. The type I errors, power (**a**) and mean squared error (MSE) of the causal effect estimates (**b**) were evaluated using 6 replicates for each of 1083 PheWAS code based phenotypes under a significance threshold of 0.05. We compared the power for *SMILE-2* against the standard two stage regression using unrelated parents (*IND-FE*) from each family. Results are shown for different pollutants and are stratified by the disease prevalence ($\hat{K}$). Combined results for all diseases are also shown. The error bars represent the standard error (SE) across simulation replicates performed under different scenarios.

the estimated heritability and community-level environmental effects are robust to mis-specified correlation structure of spatial effects, similar to what was observed for heritability estimates in standard linear mixed models[30].

We also investigated whether the length of enrollment of study participants influences our phenotype definitions and the estimates of variance components in the SMILE model (see Supplementary

Methods for more details). In brief, we compare the variance components estimates using families enrolled 6-7 years (149,710 families) and using families enrolled for 10–12 years (39,247 families) for all 1083 phenotypes. Overall, we observed a strong correlation for all variance components (Supplementary Fig. 7 and Supplementary Data 4) suggesting reduced length of enrollment has minimal impact on variance component estimates in the MarketScan data.

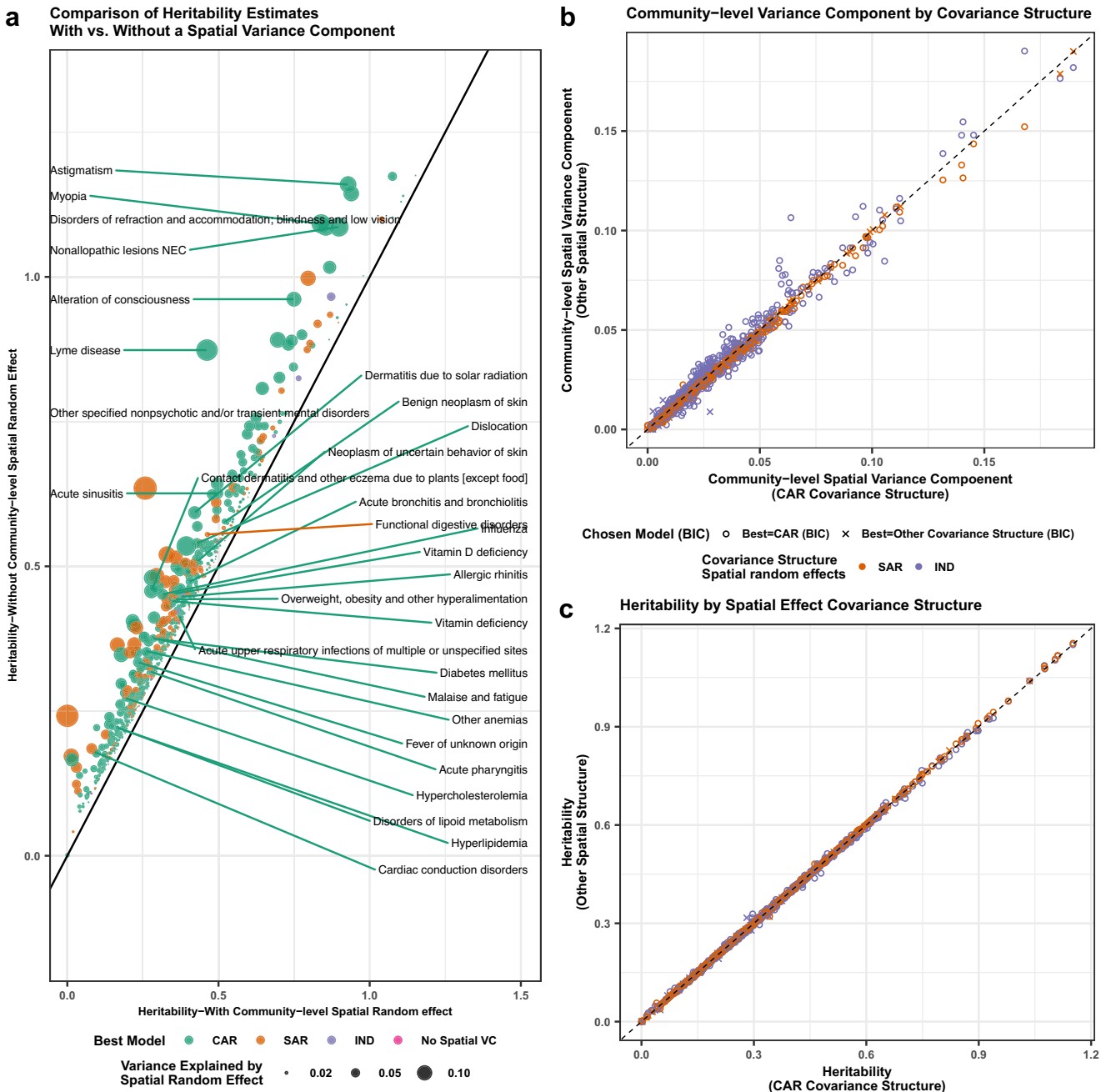

**Fig. 3 | Comparison of estimated heritability with and without accounting for community effects.** In **a**, we labeled the top 30 phenotypes with largest likelihood ratio increase after the addition of a community-location level spatial variance component. As expected from the theory, heritability estimates decrease on average by 4.3% after accounting for shared community-level environment. The models with the best BIC values are labeled and the size of the data points represents the magnitude of community-level environmental variance. In **b** and **c**, we compared variance estimates under independent normal distribution (IND), conditional autoregressive (CAR), and simultaneous autoregressive (SAR) covariance structures. We showed that regardless of the spatial covariance matrix used, (**b**) the spatial variance explained is very similar and (**c**) the estimated heritability is nearly identical.

## Landscape of heritability and community-level environment

We further stratified the diseases into 16 categories based upon their biological functions as designated by the PheWAS code mapping[13] (Fig. 4). We examined the distribution of heritability and explainable community-level spatial variation for traits within each category.

We highlighted the diseases with the largest genetic and community-level environment variance components in each category (Fig. 4A, B). Hematopoietic traits and congenital anomalies had the highest average heritability compared to other trait categories, which are concordant with other genetic studies on these traits[31]. Traits with the highest community location-level spatial variance components included diseases related to parasitic infections (e.g., Lyme disease), and allergic reactions (e.g., contact dermatitis due to plants and dermatitis due to solar radiation). For some of these diseases, the connections to spatial environment and community are clear. For example, the areas with a high incidence of Lyme disease occur primarily in the upper Midwest and northeastern regions of the United States that are more rural[32]. Allergic reactions may be triggered by air pollution or pollen, both of which are spatially correlated and can be captured by spatial random effects[33].

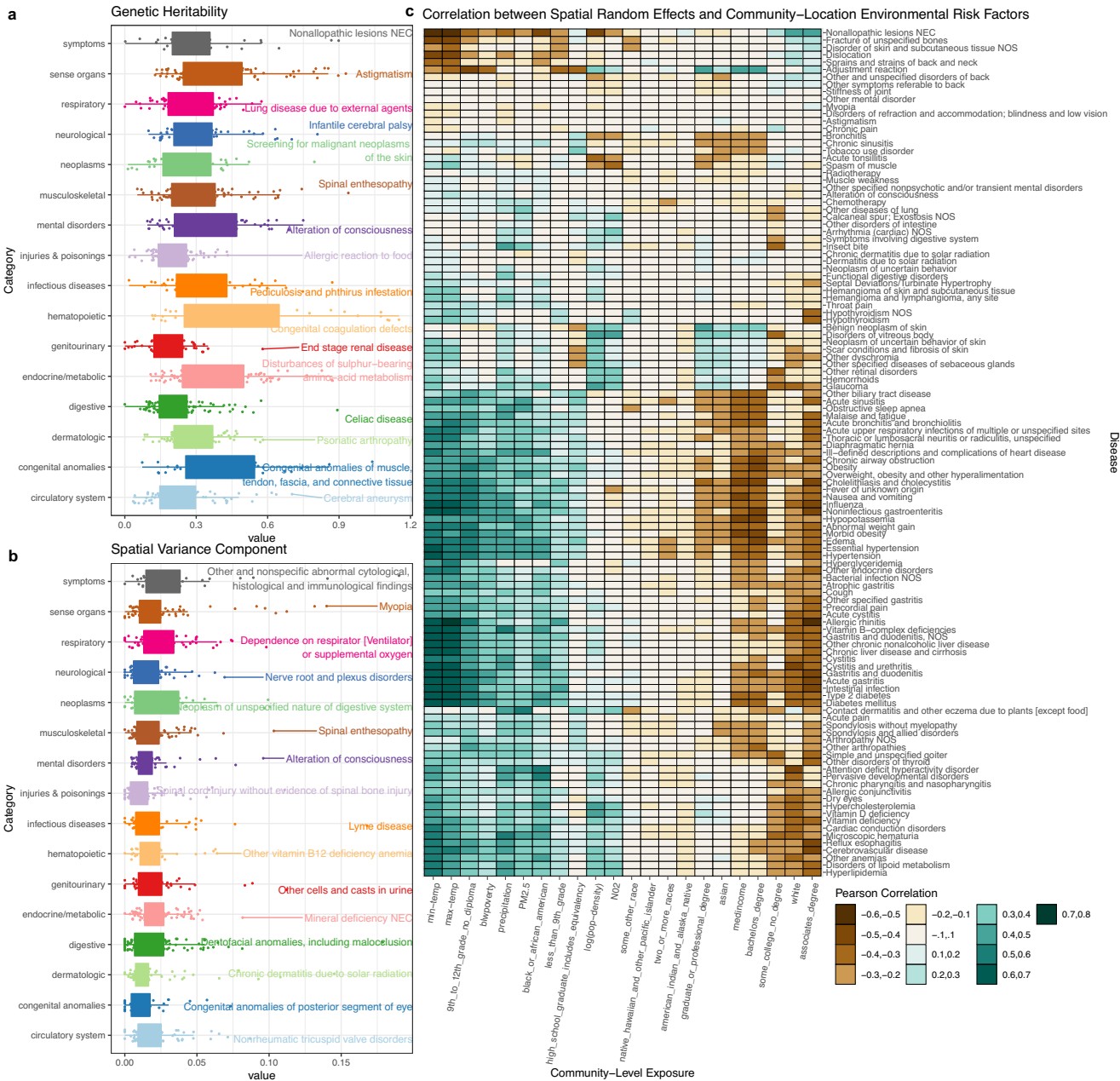

**Fig. 4 | Distribution of genetic and community effects by phenotype category.**
Panels **a** and **b** show the distribution of heritability and community-level environmental variance component in each category of PheWAS disease code and the disease with the largest variance components is labeled for each category. Minima and maxima values (excluding outliers) are represented by the lower- and upper-bound of the whiskers. Median value is represented by the bold line in the middle. First and third quartiles are represented by the lower- and upper-bound of the box. Panel **c** displays heatmap of correlations between the best linear unbiased predictor (BLUP) of spatial random effects and each individual community location-level environmental exposure. Different colors in the plot indicate different ranges of Pearson correlation coefficient values. The number of phenotypes in each categories are the following: symptoms ($n=37$), sense organs ($n=91$), respiratory ($n=70$), neurological ($n=63$), neoplasms ($n=43$), musculoskeletal ($n=91$), mental disorders ($n=58$), injuries & poisonings ($n=75$), hematopoietic ($n=40$), genitourinary ($n=55$), endocrine/metabolic ($n=100$), digestive ($n=117$), dermatologic ($n=77$), congenital anomalies ($n=38$), and circulatory system ($n=79$).

We compared our heritability estimates for diseases defined by PheWAS codes to the heritability estimates from several previously published studies (Table 2):

1. An independent study using MarketScan database ***MS1***[14], but analyzed without modeling the shared community-level environment;
2. A study that repurposed EHR data from the New York State ***NY***[45];
3. A study ***CaTCH***[3] that analyzed twins from EHR data to estimate genetic and environmental contributions, and

4. Heritability estimates based upon GWAS summary statistics from UK Biobank (**LDSC-UKB**[34]).

The *MS1*, *NY*, and *CaTCH* studies are family-based and estimate narrow-sense heritability while LDSC-UKB estimates chip heritability. We found the SMILE estimates are significantly correlated with published studies, but generally yielded smaller estimates of heritability than the GPC model and the other family-based studies, i.e., *NY*, *MS1*, and *CaTCH*. This is consistent with our simulation results, indicating

**Table 2 | Correlations of SMILE results with heritability estimates from published studies**

| Cohort | Scenario | N | H² correlation between SMILE and published study (P-value) | | Percentage of traits with overlapping conference Interval | | Mean squared differences between SMILE and published estimates of H² | | Median difference in H² (SMILE – published estimates) | |
|---|---|---|---|---|---|---|---|---|---|---|
| | | | GPC | SMILE | GPC | SMILE | GPC | SMILE | GPC | SMILE |
| CaTCH[3] | All | 540 | 0.11 (0.014) | 0.12 (0.0040) | 0.43 | 0.42 | 0.060 | 0.053 | 0.002 | −0.031 |
| | K > 1% | 405 | 0.19 ($8.7 \times 10^{-5}$) | 0.21($2.3 \times 10^{-5}$) | 0.42 | 0.41 | 0.049 | 0.044 | −0.013 | −0.045 |
| LDSC-UKB[34,79]** | All | 68 | 0.18 (0.14) | 0.16 (0.20) | 0.75 | 0.79 | 0.11 | 0.099 | 0.189 | 0.169 |
| | K > 1% | 44 | 0.33 (0.029) | 0.30 (0.050) | 0.66 | 0.73 | 0.12 | 0.098 | 0.205 | 0.169 |
| MS[14] | All | 63 | 0.79 ($1.7 \times 10^{-14}$) | 0.83 ($2.7 \times 10^{-17}$) | 0.44 | 0.26 | 0.016 | 0.020 | −0.062 | −0.101 |
| | K > 1% | 52 | 0.76 ($5.77 \times 10^{-11}$) | 0.79 ($4.7 \times 10^{-12}$) | 0.41 | 0.2 | 0.017 | 0.021 | −0.064 | −0.106 |
| NY[15] | All | 33 | 0.57 (0.00051) | 0.52 (0.0015) | 0.7 | 0.65 | 0.029 | 0.032 | −0.009 | −0.049 |
| | K > 1% | 31 | 0.48 (0.0069) | 0.48 (0.0069) | 0.68 | 0.68 | 0.028 | 0.030 | −0.009 | −0.053 |

(K > 0.01 indicates traits with an observed prevalence greater than 1%.) All p-values are for two-sided hypothesis tests.
**The NY and LDSC-UKB studies are based on ICD-9/10 codes. It is difficult to directly compare ICD code defined and PheWAS code defined phenotypes. For this reason, we restricted our comparisons to ICD-9/10 codes which could be mapped to a unique PheWAS code.

the shared community-level environmental risk, when left unaccounted for, could add to the upward bias in the heritability estimates from family-based studies. More details on the comparison can be found in the Supplementary Methods.

For diseases with strong environmental contributions, our SMILE model likely offers much refined estimates of heritability. For example, our heritability estimates for type 2 diabetes (T2D) decreased from 37.7% to 28.4% after accounting for spatial community effects. Even after redefining T2D cases using both ICD diagnostic codes and T2D medication codes[35] (see Supplementary Methods for more details), the heritability estimate only increased to 31% (Supplementary Data 5-6). Both estimates are lower than a majority of previous estimates from family studies of T2D, i.e., several previous studies have produced heritability estimates in the range of (0.26–0.69)[36–40]. On the other hand, the result is more concordant with a recent large-scale study analyzing UK Biobank participants[34] which consists of primarily unrelated individuals. It obtained heritability estimates ranging from 19.6% to 33.2% depending on model specification using whole-genome data and whether rare or low-frequency SNPs were included for estimation[41]. Along similar lines, our heritability estimate for obesity decreased from 53.1% to 46.3% when adjusted for spatial community effects, while classical twin studies reported estimates as high as 70%[42]. The spatial community-level random effects for T2D showed a strong correlation with those for obesity ($\hat{\rho} = 0.67$), and a number of other lipid metabolism-related traits (i.e., Hyperlipidemia $\hat{\rho} = 0.75$, Hypercholesterolemia $\hat{\rho} = 0.61$). Obesity is well known to be the number one leading risk factor for T2D[43]. The correlations in community-level environment effects underscore the well-known shared etiology of T2D and obesity attributable to environmental factors[44,45], and increase our confidence in the validity of the effects captured by the community-level spatial variance component.

### Extensive correlation between spatial random effects and environmental risk factors

Spatial random effects can capture a wide range of environmental risk factors, including many that are not often measured or controlled for in genetic studies. To gain a better understanding of what underlies our estimates of community-level risk, we integrated potential community-level environmental risk factors (CLERF) from external data sources into the MarketScan dataset according to the county or MSA locations. The additional CLERF variables include averaged minimum and maximum monthly temperature and precipitation levels, averaged $PM_{2.5}$ and $NO_2$ air pollution, as well as sociodemographic variables of median income, population density, poverty rates, education levels, and racial distributions at the county or MSA level from the 2015 ACS community survey[18] (Supplementary Data 7-8). We calculated the total

community-level environmental contribution for each disease at each MarketScan location using the best linear unbiased predictor (BLUP) of the spatial random effect from the SMILE model. As the external CLERF variables are not included as covariates in the SMILE model, we regressed the BLUPs over these risk factors to assess their impact.

We calculated the correlation between CLERF and BLUPs for 115 diseases with a prevalence of at least 2% and with an estimated spatial variance of at least 2% (Fig. 4C). For a majority of diseases, we observed that increased disease risk is correlated with indicators of lower socioeconomic status (SES), such as lower median income, the percentage of individuals with high school as the highest education, and poverty rate. Examples of lower SES-associated diseases included obesity, diabetes, chronic liver disease, chronic obstructive pulmonary disease (COPD), influenza, and fever. Interestingly, several traits were observed to be associated with higher SES, including benign neoplasms of the skin, hemorrhoids, and adjustment reaction (a more severe reaction than expected following a stressful event). We speculate that these findings may be attributable to disparities in education and access to healthcare for lower SES groups. For example, previous research has found that low SES is associated with more advanced melanoma at diagnosis, and that individuals with lower SES were less likely to be concerned about melanoma risks, or seeking screening and treatment by their physicians[46,47], explaining why higher SES groups would be more likely to have higher reported neoplasm incidences. Multiple prospective studies have noted that low SES is associated with poor mental health outcomes following stressful events[48,49]. However, just as other observation studies, our sample is an observational scan of EHR databases, the inverse relationship we observe in our study between SES and adjustment reaction may be due to ascertainment. Similar explanations may underlie the association between higher SES and hemorrhoids, which has been noted in previous studies[50].

### Estimating causal effects of air pollution across 1083 phenotypes

We used SMILE-2 to assess the causal effects of $PM_{2.5}$ and $NO_2$ air pollution on 1083 PheWAS disease codes with an observed prevalence of at least 0.1%.

The distribution for satellite-inferred estimates of $PM_{2.5}$ and $NO_2$ air pollution at the centroids of each MarketScan location were shown in Supplementary Figs. 1–3 (Methods), which was based upon averages across all years. The long-term averaged wind speed and direction information at each location were shown in Supplementary Fig. 8, which were used as instrumental variables for pollution to reduce the correlation between pollution levels and any unobserved confounding effects (Methods).

For each disease, we first applied SMILE-2 to analyze $PM_{2.5}$ and $NO_2$ air pollutants, and then also analyzed a constructed total pollutant level based upon the sum of the standardized $PM_{2.5}$ and $NO_2$ levels ($\tilde{P}_{SUM}$). After Bonferroni correction, we found 135/1083 (12.5%) of the phenotypes to have a significant association with either $PM_{2.5}$ (Fig. 5A), $NO_2$ (Fig. 5B), or $\tilde{P}_{SUM}$ (Fig. 5C) air pollution. The estimated causal effect is positive for 105 of 135 significant traits (77.8%), indicating elevated pollution levels increase disease risk.

Among the significant causal effects, we found that the two pollutants affect different classes of diseases. For example, diseases significantly associated with $PM_{2.5}$ but not with $NO_2$ included multiple sleep disorders (hypersomnia ($\widehat{OR}_{PM_{2.5}} = 1.13, P = 5.4 \times 10^{-17}$), obstructive sleep apnea ($\widehat{OR}_{PM_{2.5}} = 1.04, P = 1. \times 10^{-8}$), parasomnia ($\widehat{OR}_{PM_{2.5}} = 1.07$, $P = 1.2 \times 10^{-6}$), narcolepsy ($\widehat{OR}_{PM_{2.5}} = 1.13, P = 5.3 \times 10^{-9}$)), respiratory infections[51] (acute sinusitis ($\widehat{OR}_{PM_{2.5}} = 1.07, P = 3.8 \times 10^{-24}$), acute bronchitis and bronchiolitis ($\widehat{OR}_{PM_{2.5}} = 1.05, P = 1.1 \times 10^{-18}$)), ear infections[52] (otitis media ($\widehat{OR}_{PM_{2.5}} = 1.04, P = 1.2 \times 10^{-23}$), and attention deficit hyperactivity disorder (ADHD)[53] ($\widehat{OR}_{PM_{2.5}} = 1.04, P = 1.4 \times 10^{-7}$), (Fig. 5A).

At the same time, diseases associated with $NO_2$ pollution but not with $PM_{2.5}$ highlighted distinct symptomatology including multiple gastro-intestine-related disorders[54] (Gastritis ($\widehat{OR}_{NO_2} = 1.1$, $P = 4.4 \times 10^{-7}$), IBS ($\widehat{OR}_{NO_2} = 1.1, P = 8.1 \times 10^{-9}$)), as well as both type I and type 2 diabetes[55] ($\widehat{OR}_{NO_2} = 1.15, 1.17$ and $P = 2.5 \times 10^{-6}, 2.3 \times 10^{-6}$ respectively) (Fig. 5B). Additionally, we found several lipid metabolism associated diseases are causally linked with $NO_2$ (e.g., hyperlipidemia $\widehat{OR}_{NO_2} = 1.09, P = 1.3 \times 10^{-7}$ and hypercholesterolemia $\widehat{OR}_{NO_2} = 1.10$, $P = 7.5 \times 10^{-6}$). This is concordant with discoveries from several previous studies in the Chinese populations[56,57], with other research indicating that $NO_2$ may play a role in the regulation of lipid metabolism and may promote the formation of fatty plaque in arteries[58–60]. Compared to $NO_2$, $PM_{2.5}$ may have a more damaging direct effect on lung function, and it has been shown that $PM_{2.5}$ can cause inflammation and a weakened immune-system defense, leaving the respiratory system prone to infection[61].

We also rediscovered associations with diseases of low prevalence, which have primarily only been studied in relation to air pollution for specific subpopulations. For example, previous studies showed that cystic fibrosis patients exposed to high levels of $PM_{2.5}$ air pollution are more likely to develop methicillin-resistant Staphylococcus aureus (MRSA), which is an antibiotic-resistant infection[62]. In our study, we recapitulate the causal relationship between MRSA and $PM_{2.5}$ in the general population as well ($\widehat{OR}_{PM2.5} = 1.05, P = 5.1 \times 10^{-6}$). The causal effect of $PM_{2.5}$ on MRSA in the general population underscores the important link between air pollution and infectious diseases from a public health perspective.

## Discussion

In this article, we develop the SMILE model to jointly quantify the contributions of genetics and correlated community level shared environment on disease phenotype variation. We applied the method to analyze insurance claim data using the MarketScan dataset with more than 50 million individuals. We refined the estimates of genetic heritability and community-level environmental variance components. We also quantified the causal effects of air pollutants $PM_{2.5}$ and $NO_2$ for 1,083 diseases.

The refined heritability estimates by SMILE may help reconcile the discrepancy between heritability estimates from family studies and GWAS using unrelated individuals[63], as it helps correct for the upward bias induced by the correlated community-level environment in family-based variance components models. SMILE does not need genotype data as input, making it uniquely suitable for analyzing insurance claim data without genetic information. It also differs from genome wide interaction studies (GWIS), which uses explicitly measured environmental variables and genetic information to identify genetic variants interacting with environment. As it is virtually

impossible to measure all environmental risk factors to the trait variation, GWIS may share the same limitations of GWAS where unmeasured environmental risk factors may confound heritability estimates. GWIS is also not applicable to insurance claim data as it needs genotype information. In contrast, SMILE captures the contributions of spatially correlated environmental risk factors to the trait variation without having to explicitly measure each environmental risk factor individually. Thus, SMILE complements GWIS and is essential for deriving more accurate variance components estimates in the presence of unmeasured environmental exposures.

Our comprehensive catalog of heritability estimates derived from EHR-based phenotypes offers a unique reference to quantify environmental contribution and assess the "missing heritability" for complex diseases. To ensure the validity of our results, we have conducted comprehensive robustness analyses and simulations that suggest our results are robust against different phenotype definitions, misspecified pedigrees, and measurement errors in wind/pollution levels. These robustness analyses ensures the usefulness of SMILE and its extensions to insurance claim data and national EHR-based biobanks, such as UK Biobank[64] and *All of Us*[65].

Another contribution of our work is that we showed that different pollutants in the air may have distinct causal effects on different diseases. This contrasts many epidemiological studies, where different sources of ambient air pollution (i.e., $PM_{2.5}$, $NO_2$, ground-level ozone, and carbon monoxide) are combined into an aggregate measure of air quality, not distinguishing the specific mechanisms by which individual pollutants impact disease. These results would be useful for generating hypotheses for follow-up analyses.

There are several aspects of our analyses that warrant discussion. First, MarketScan database only includes individuals with employer sponsored insurance policies. As such, low-income families may not be well-represented[14]. While the results are valid for the population the dataset represents, it is important to exercise caution when extrapolating our findings to different populations.

Second, as an insurance claim database, MarketScan data only includes medical information during a limited period of time[14]. For example, the data on children is only up to the age of 26, which is the maximum age a child can be covered by their parent's health insurance in the US. Thus, the prevalence for late-onset conditions may be lower in children when compared to parents. We account for this by (1) limiting our analysis to families enrolled in the database for at least 6 years, (2) limiting our analysis to families where all children are at least 10 years old at the time of entry into the database, and (3) excluding families where the age at enrollment of the youngest family member was less than the 5th percentile of the age of diagnosis for the phenotype of interest and (4) including age and $age^2$ as covariates in both SMILE and SMILE-2 to account for the impact of age. Despite the limitation of the datasets, our heritability estimates are comparable to estimates obtained from other data types (Table 2), which demonstrate the effectiveness of our filtering criteria and the validity of our results.

Third, EHR-derived phenotypes may not be completely accurate. For example, substance abuse disorder cases may be underrepresented if a substantial proportion of those affected do not seek medical treatment. For some diseases, the presence of a medical diagnostic code may highlight the differences in healthcare-seeking behavior rather than a true representation of disease prevalence. Poor coding practices for various traits may have negative net impacts on public health research, and some research has provided evidence that EHR documentation is heterogeneous across medical providers, practices, and physicians, including the documentation of diagnostic codes[66]. In this regard, spatial random effects may be viewed as a potentially effective way of controlling for these biases, similar to the use of a linear mixed model in GWAS to account for unexplained population structure[67].

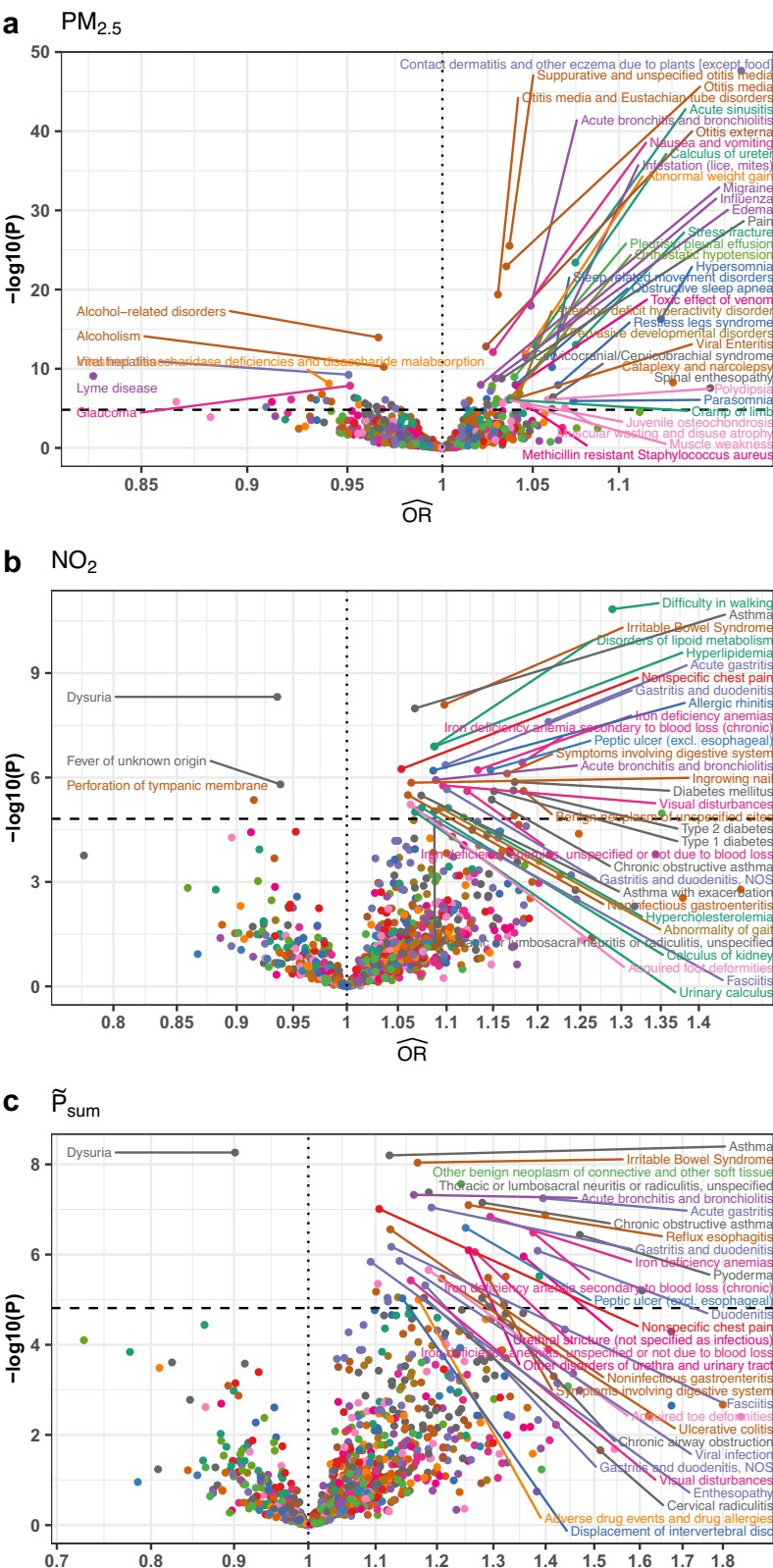

**Fig. 5 | Volcano plots of causal effect estimates from SMILE-2 model for pollution on diseases.** We plotted the estimated odds ratios against −log₁₀(*p*-values) for the causal effects of (**a**) PM₂.₅, (**b**) NO₂, and (**c**) the sum of the two pollutants ($\widetilde{P}_{SUM}$). All *p*-values are for two-sided hypothesis tests and are unadjusted for multiple comparisons. Dashed horizontal line represents Bonferroni significance threshold for testing 1083 diseases and three pollutants.

Fourth, causal inference results should be carefully interpreted. The Bradford-Hill criterion[68] is a standard benchmark for assessing causality in epidemiological studies, and should be considered for all causal inferences. Temporality is one assumption, which states that the pollution exposure precedes the disease. Here, we used a single long-term average of $PM_{2.5}$ and $NO_2$ air pollution at MarketScan participant locations as the measurement of pollution exposure[69]. Therefore, the analysis implicitly assumes that the average level of pollution is representative of the pollution exposure for individuals at each location. While this assumption is valid during the period where our data was collected, environmental factors may experience transient changes. For example, during the COVID-19 pandemic, $NO_2$ levels were significantly reduced but $PM_{2.5}$ levels remained similar to other times. This is because $NO_2$ is emitted during fuel combustion of all motor vehicles and airplanes, but $PM_{2.5}$ is mainly emitted by diesel commercial vehicles and remained largely unchanged by quarantine restrictions[70]. Understanding how the transient changes in individual pollutants impact diseases in future works may lead to more effective and better-informed environmental policies and air quality regulations.

When causal effects are observed, it is also helpful to further assess potential explanations and biological feasibility. We observed several significant associations where air pollution was negatively associated with diseases that potentially originate from sexually transmitted infections (painful urination, viral hepatitis) (Fig. 5). This correlation is consistent with increased sexually transmitted infections across rural areas in the U.S. compared to more heavily polluted regions, which may stem more directly from the lack of access to public health resources and social conservatism[71].

We also envision several exciting areas for future research which exploit the pedigree structures, deep phenotyping, and massive sample sizes of EHR datasets. For one, computational constraints often require dichotomizing diseases into binary traits indicating whether the trait was "ever" or "never" observed for a patient, yet EHR records are inherently longitudinal in nature. Modeling strategies that can account for time-to-event outcomes and/or recurrent events (such as common colds, broken bones, or infections) may yield greater insights into the etiology of certain diseases. Similarly, modeling air pollution changes over time could also yield additional findings. Incorporating spatial random effects and modeling correlated community-level environment could also lead to applications outside the scope of this paper, e.g., improving the power of genetic association tests in national biobanks[72].

Together, our methods for modeling spatially dependent community-level environmental risk open new venues to analyze national biobanks and explore the genetic architecture of complex traits. Our improved estimates of heritability, environmental contribution, and causal effects for air pollution across the phenome offer a valuable foundation upon which future studies may be built.

## Methods

Here we describe the SMILE model for quantifying the genetic and community-level environment contribution. The extension of the SMILE-2 model for assessing the causal effect for air pollution, the description of the datasets used, and the simulation analyses are left to the Supplementary Methods.

### SMILE model of genetic heritability, family environment, and community location random effects

We developed the SMILE model to jointly characterize the genetic, family-level, and community-level environmental variance components. We also refer to the model as GPC + S according to the variance components included. We included age, the number of months

enrolled in the dataset, and the indicator variables for sex, and the first year of enrollment as individual-level fixed-effect covariates. A total of $N_F$ nuclear families (with $N$ individuals) were used in the analysis. The full SMILE model (i.e., GPC + S) may then be specified as

$$Y = X\pi + u_g + Z_s u_s + Z_{par} u_{par} + Z_{child} u_{child} + \epsilon \qquad (1)$$

where $Y = (y_1, \ldots, y_N)$ is the vector of case-control status. To facilitate the presentation of the method, we assume that $Y$ is arranged by families, and within each family, the phenotypes are arranged in the order of father, mother, and children. $X$ denotes the design matrix for the fixed effect individual-level covariates with effect $\pi$. $u_g, u_{par}, u_{child}$, and $u_s$ are respectively the genetic, shared parental, and children's environmental random effects, and community-level spatial random effects. $Z_s, Z_{par}$, and $Z_{child}$ denote the indicator matrices, mapping each individual to their corresponding random effects in $u_s, u_{par}$, and $u_{child}$.

More specifically, $u_{par} = (u_{par,1,1} \ldots, u_{par,N_F,1}, u_{par,N_F,2}, \ldots)$ and $u_{child} = (u_{child,f,1}, \ldots, u_{child,N_F,1}, u_{child,N_F,2}, \ldots)$ are vectors of independent and identically distributed normal random variables. In family $f$, the parents share random effect $u_{par_{f,1}}$ and the children have random effects $u_{par_{f,2}}, u_{par_{f,3}}$, etc. Similarly, in family $f$, children share random effects $u_{child_{f,3}}$, while parents have random effects $u_{child_{f,1}}$ and $u_{child_{f,2}}$, as children and parents may have different environmental exposures.

Within each family, the correlation between genetic random effects is determined by kinship matrix $G$. In the example of a quad family (nuclear family with 2 children), the kinship matrix is given by:

$$G = \begin{pmatrix} 1 & 0 & 0.5 & 0.5 \\ 0 & 1 & 0.5 & 0.5 \\ 0.5 & 0.5 & 1 & 0.5 \\ 0.5 & 0.5 & 0.5 & 1 \end{pmatrix} \qquad (2)$$

where the first two rows represent parents and the last two rows represent children in the family. Each entry in the matrix represents the genetic kinship between corresponding individuals in the family.

The genetic random effects across all families satisfy:

$$u_g \sim N\left(0, \sigma_g^2 \, blk\,diag\left(G_1, \ldots, G_{N_F}\right)\right)$$

where $blk\_diag$ represents block diagonal matrices.

The community-level spatial random effect $u_s = (u_{s,1}, \ldots, u_{s,L})$ is a vector of length $L$, with individuals located in location $l$ having random effect $u_{s,l}$ and $L$ being the number of unique MarketScan county or MSA locations). To model the spatial dependence between families, we considered conditional autoregressive (CAR), simultaneous autoregressive (SAR), and independent (IND) covariance matrices for community-level spatial random effects, as they cover a range of scenarios, and it is computationally feasible to apply them to large datasets. Specifically, under CAR, SAR or IND models, the spatial random effects follow:

$$\text{CAR} : u_s \sim \text{MVN}(0, \Sigma_{CAR}) \qquad (3)$$

$$\text{SAR} : u_s \sim \text{MVN}(0, \Sigma_{SAR}) \qquad (4)$$

$$\text{IND} : u_s \sim \text{MVN}(0, \sigma_s^2 I) \qquad (5)$$

To describe the covariance matrix of the spatial random effects (i.e., $\Sigma_{CAR}$ and $\Sigma_{SAR}$), we first define the weight matrix $W$ as a $L \times L$ symmetric matrix. $W$ has diagonal entries of 0 and off-diagonal entries of 1 for pairs of locations that share a common border. MSA's were

considered as 'sharing borders' with the counties they encompass. Adjacencies between counties were identified using the R package *spdep*[73]. $\mathbf{W}_+$ is obtained from $\mathbf{W}$ by standardizing its rows, so that the entries from each row add up to 1 (according to the definition of $\mathbf{W}$, the normalizing factor $\sum_j W_{ij}$ equals the number of 'neighbors' to location $i$). $\mathbf{M}$ is a diagonal matrix with diagonal elements $M_{ii} = \left(\sum_j W_{ij}\right)^{-1}$.

Using the defined weight matrices, the covariance matrices for CAR and SAR models are specified as

$$\mathbf{\Sigma_{CAR}} = \sigma_s^2 \left(\mathbf{M}^{-\frac{1}{2}}\left(\mathbf{I} - \rho\mathbf{M}^{-\frac{1}{2}}\mathbf{W}_+\mathbf{M}^{\frac{1}{2}}\right)\mathbf{M}^{-\frac{1}{2}}\right)^{-1} \tag{6}$$

$$\mathbf{\Sigma_{SAR}} = \sigma_s^2 \left(\left(\mathbf{I} - \rho\mathbf{M}^{-\frac{1}{2}}\mathbf{W}_+\mathbf{M}^{\frac{1}{2}}\right)\mathbf{M}^{-1}\left(\mathbf{I} - \rho\mathbf{M}^{\frac{1}{2}}\mathbf{W}_+^{\mathbf{T}}\mathbf{M}^{-\frac{1}{2}}\right)\right)^{-1} \tag{7}$$

The covariance matrices for CAR and SAR models have unequal diagonal elements. For a given estimated parameter $\sigma_s^2$, the spatial random effects explain different amounts of phenotypic variance for individuals at different locations. In order to better quantify the phenotypic variance explained by spatial random effects, we calculate a Gower factor[74–76]

$$\tilde{\sigma}_s^2 = \frac{tr\left[\mathbf{Z_s}\left(\mathbf{I} - \frac{1}{N}\mathbf{11}^{\mathbf{T}}\right)\mathbf{\Sigma_S}\mathbf{Z_s^T}\right]}{N-1} \quad 8$$

The Gower factor can be considered as the averaged variance of spatial random effects across individuals.

For all traits, we report $\tilde{\sigma}_s^2$ as the phenotypic variance contributed by spatially-correlated community-level environment.

### Conversion of variance components from observed scale to liability scale

All disease outcomes are binary. We use linear regression models to estimate variance components on the observed scale. This has been a widely used approach in human genetics and is computationally efficient compared to generalized linear mixed models for large datasets[77]. To facilitate the comparison of estimates across diseases with different prevalence, we will convert them to liability scale. The details for the conversion are given in the Supplementary Methods. As we demonstrate in Results and Fig. 1, the conversion yields unbiased results across different scenarios.

### Fitting the model with laplace approximation

We make extensive use of the R package TMB[78] to estimate model parameters, which relies on Automatic Differentiation software to calculate the gradients of the objective function obtained by Laplace approximation. More details can be found in the Supplementary Methods.

### Two stage regression model for causal inference of PM$_{2.5}$ and NO$_2$ air pollution

We extend SMILE model into a two-stage regression model SMILE-2 for assessing the causality of air pollution levels using wind speed and direction as instrument variables. Details are shown in the Supplementary Methods.

### Ethical approval

This study is deemed non-human subject research and approved by Penn State College of Medicine IRB.

### Reporting summary

Further information on research design is available in the Nature Portfolio Reporting Summary linked to this article.

## Data availability

We provide all the data that support the findings of this study in this published article (and its supplementary information files). The raw data from Truven MarketScan are available for licensed users. A user license could be obtained by following the instructions at https://marketscan.truvenhealth.com/marketscanportal/. Multiple external datasets of community-level risk factors were incorporated in this study. This included demographic data from the 2015 American Community Survey 5-year estimates[18], satellite-derived measurements of air pollution including PM$_{2.5}$[19,20] and NO$_2$[21,22], and wind direction and wind speed data[23].

## Code availability

The software package implementing the SMILE and SMILE-2 model is available at https://github.com/dan11mcguire/smile and the linked Zenodo repository (https://doi.org/10.5281/zenodo.11081928).

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

## Acknowledgements

Some of the materials employed in this work were provided by the Center for Applied Studies in Health Economics (CASHE) at Penn State University College of Medicine. H.M. is funded by Computation, Bioinformatics, and Statistics (CBIOS) NIH-sponsored Research Training Grant (5T32GM102057-10) and NIH F30 Ruth L. Kirschstein National Research Service Award Individual Predoctoral MD/PhD Fellowship Award by the National Institute of General Medical Sciences (F30GM151848). D.J.L. is also supported by NIH grants R01ES036042, R01HG011035, and R01AI174108 and by the Artificial Intelligence and Biomedical Informatics pilot funding program from the Penn State College of Medicine.

## Author contributions

D.M., D.J.L., and B.J. conceived the study. D.M. and H.M. led the data analysis. D.M., H.M., L.Y., J.X., and A.M. conducted analyses. A.B., Q.L., L.C., D.J.L., and B.J. helped with data interpretation. D.M., H.M., D.J.L., and B.J. prepared the manuscript. All authors contributed to manuscript editing and approved the manuscript.

## Competing interests

The authors declare no competing interests.
