## [Peer Review File · Nature Communications]

Dissecting Heritability, Environmental Risk, and Air Pollution Causal Effects Using >50 Million Individuals in MarketScanREVIEWER COMMENTS

Reviewer #1 (Remarks to the Author):

This is an interesting large-scale study that integrates diagnosis data with real-world environmental exposure data. It is in many ways an impressive undertaking given the size of the data set and the way these data are integrated in one statistical model. I am however not sure that the approach and the quality of the data allows for producing accurate and superior heritability estimates as the paper indicates. Several factors contribute to this view, for example that the paper analyzes noisy nuclear families (deidentified data) who were enrolled in the database for as little as 6 years between 2005 and 2017. There is no robustness estimate provided, comparing families being enrolled for 10-12 years to those enrolled for 6-7 years for example. This weakens the link to the exposure data, which otherwise in principle appears to be of high quality.

I am also not sure the conversion of variance components from observed scale to liability scale is robust. To me it appears questionable that the conversion of estimates from the observed scale to the liability works without introducing bias. It would be advisable to further support the assumptions inherent in the analysis through a more thorough theoretical and/or a simulation study.

The paper addresses numerous diseases, but there is no thorough analysis of the validity of the diagnoses in the data set and hence of the heritability estimates across diseases. The paper presents type 2 diabetes as a case and reports a lower heritability estimate than several previous studies. It is well known that using diagnosis codes from EHRs for type 2 diabetes is quite unreliable as diabetes is handled across primary and secondary care settings. This leads often to wrong estimation of incidence and quite noisy estimates for time of onset. One typical strategy is to combine data on diagnoses with information on glucose lowering medications from redeemed prescriptions, and sometimes various procedure codes. In the paper the authors rely on diagnosis codes only. This could be problematic in an analysis that aims for relating causal factors to each other. I am not sure that the incomplete data across primary and secondary care settings in the MarketScan data allow for systematically establishing whether the diagnoses are sufficiently correct and whether patients actually are treated or not (in order to provide better heritability estimates than previous papers). IBM's website also provides disclaimers, ibm.com/downloads/cas/OWZWJ0QO :

"MarketScan databases are based on a large convenience sample. Because the sample is not random, it may contain biases or fail to generalize well to other populations. However, these data can complement other data sets or be used as benchmarks against them. – Data come mostly from large employers; medium and small firms may be underrepresented, although the MarketScan Research Databases include a large amount of data contributed from health plans."

Early onset disease versus late onset is also not a theme in the paper. It would be relevant to ask why this is not important in the context of the work presented.

I think the strength of the paper lies in the integration and handling of the exposure data, but I am not convinced that the statistical model, given the noisy definition of families and the variable validity of the diagnosis registration, generally can lead to a more accurate estimate of heritability compared to previous papers as the work states.

Reviewer #2 (Remarks to the Author):

GENERAL COMMENTS

McGuire et al. conducted a simulation study examining how combining genetic, familial, and environmental variance components may help to better estimate genetic heritability of air pollution.

The study utilized data from several different resources, including electronic health records (EHR) was then linked with publicly available estimates of environmental data (including PM2.5 and NO2). A spatial mixed linear effect model (SMILE) was used to estimate heritability while incorporating both genetics and environmental variance components. The simulation study concluded that models including environmental variance parameters in the SMILE model provided better estimates of genetic heritability of air pollution. The study also conducted an instrumental variable analysis to estimate putative causal effect of PM2.5 and NO2 on various health outcomes. A couple suggested associations for variants located in the FAT3 and NLRP3 loci were identified. The manuscript is overly complex and ambitious. However, the ambitious nature of the manuscript with the inclusion of several advanced methodologies detracts and hinders the manuscript's clarity, interpretation and overall motive. Several methodological decisions made were not clarified and/or substantiated. Additional justification for employing the methods described is severely needed (especially for the instrumental variable analysis). Additional comments are provided below.

SPECIFIC COMMENTS

Introduction

The authors mention the issues with traditional genome wide association studies (GWAS) but fail to recognize the potential benefit of genome wide interaction studies (GWIS) in examining the interaction between both genetics and environmental factors on identifying associated loci.

Introduction

The use of the EHR database could have severe selection bias issues which were never addressed or mentioned in the paper.

Results

The manuscript further fails to account for potential measurement error issues in using publicly available estimates for pollution measures and applying them to individuals. Furthermore, it is possible that the instrumental variable used in the analysis themselves are measured with error.

Statistical Methods Overview

It is unclear what the children-level family environment random effects represent, and how it would differ from parental level effects given nuclear families were used and I believe assume to have the same level of exposure.

Statistical Methods Overview

The authors need to further clarify how the variance for the estimates in the model sampled with replacement was adjusted to account for the replacement of families.

Estimating Causal Effects

The authors did not provide any assumptions that are made in an instrumental variable analysis and no evaluation regarding the instrumental variables and whether or not they meet the strict assumptions of an IV were conducted. The authors should present confidence intervals for their estimates.

REVIEWER COMMENTS

Reviewer #1 (Remarks to the Author):

1. This is an interesting large-scale study that integrates diagnosis data with real-world environmental exposure data. It is in many ways an impressive undertaking given the size of the data set and the way these data are integrated in one statistical model. I am however not sure that the approach and the quality of the data allows for producing accurate and superior heritability estimates as the paper indicates. Several factors contribute to this view, for example that the paper analyzes noisy nuclear families (deidentified data) who were enrolled in the database for as little as 6 years between 2005 and 2017. There is no robustness estimate provided, comparing families being enrolled for 10-12 years to those enrolled for 6-7 years for example. This weakens the link to the exposure data, which otherwise in principle appears to be of high quality.

RESPONSE: Thank you for the comment! We are grateful that the reviewer finds our work “interesting” and “an impressive undertaking”. We agree it is important to assess the robustness of the estimates. As suggested, we have compared SMILE model estimates using individuals enrolled for 6-7 years with those enrolled for 10-12 years. Specifically, by restricting analysis to individuals enrolled for 6-7 years and 10-12 years, we identified 149,710 and 34,853 nuclear families respectively. The correlation of variance component estimated using families enrolled for 10-12 years and families enrolled for 6-7 years is high for diseases with prevalence > 1%, i.e., the correlation of spatial, genetic, parental and children’s variance components are 0.93, 0.95, 0.96, and 0.97 respectively. The correlation becomes slightly lower for diseases with lower prevalence. The observation is not surprising, as diseases with low prevalence have much fewer cases in the dataset. The resulting estimates can be noisier, and the correlation be lower. We have added a new section called “*Robustness Analyses*” in the Supplementary Text. We have summarized our methods and results in a sub-section called “*Impact of the length of enrollment on heritability estimates*” (Page 12 paragraph 4) and included Supplementary Tables 4 and Supplementary Figure 7 to summarize the results.

2. I am also not sure the conversion of variance components from observed scale to liability scale is robust. To me it appears questionable that the conversion of estimates from the observed scale to the liability works without introducing bias. It would be advisable to further support the assumptions inherent in the analysis through a more thorough theoretical and/or a simulation study.

RESPONSE: Thank you for the comment! We already provided theoretical derivation of the conversion from observed scale to liability scale in the Supplementary Text in section 6 called “*Conversion to Liability Scale Variance Components*” (Page 15). We also provide theoretical derivation for standard errors of liability-scale variance components in the Supplementary Text (section 7 “*Standard Errors for Liability-scale Variance Components by Multivariate Delta Method*”, Page 17). To further support the unbiasedness of the conversion, we also present simulation results in Figure 1b-c. In both panels of the figure, we show the converted heritability estimates in liability scale is unbiased.

3. The paper address numerous diseases, but there is no thorough analysis of the validity of the diagnoses in the data set and hence of the heritability estimates across diseases. The paper presents type 2 diabetes as a case and reports a lower heritability estimate than several previous studies. It is well known that using diagnosis codes from EHRs for type 2 diabetes is quite unreliable as diabetes is handled across primary and secondary care settings. This leads often to wrong estimation of incidence and quite noisy estimates for time of onset. One typical strategy is to combine data on diagnoses with information on glucose-lowering medications from redeemed prescriptions, and sometimes various procedure codes. In the paper the authors rely on diagnosis codes only. This could be problematic in an analysis that aim for relating causal factors to each other. I am not sure that the incomplete data across primary and secondary care settings in the

MarketScan data allow for systematically establishing whether the diagnoses are sufficiently correct and whether patients actually are treated or not (in order to provide better heritability estimates than previous papers).

RESPONSE: Thank you for the comment! We acknowledge the potential limitations in the phenotype definition by using only diagnostic code. For specific diseases, refined phenotyping algorithms exist, but it is infeasible to do so for each phenotype in the phenome. Using ICD9/10 codes or PheWAS codes to define phenotype is a standard practice for phenome-wide studies [1-3].

We agree that using a combination of diagnosis and glucose-lowering medications might lead to more accurate phenotype definitions for Type 2 diabetes. As suggested, we re-ran the SMILE model using prescription information and diagnostic code to define the T2D phenotype [4]. We compared the new heritability estimate with the original estimate based only on T2D diagnostic codes. Both heritability estimates were similar: Using the T2D case definition by adding prescription information, the heritability increased from 0.284 (SE=0.009) to 0.31 (SE=0.008). Previous twin and family-based studies report heritability of T2D to be 22%–73% [5]. Our estimate is within this range. Importantly, using this new phenotype definition, controlling spatially correlated community-level environment still yields lower heritability and corrects the upward bias, as what we would expect theoretically. We have added a new sub-section called “*Impact of drug code on phenotype definition and heritability estimates*” to the section called “*Robustness Analyses*” in the Supplementary Text (Page 12 paragraph 6). We have also included Supplementary Tables 5 and 6 to summarize the results.

4. IBMs website also provide disclaimers, ibm.com/downloads/cas/OWZWJ0QO: "MarketScan databases are based on a large convenience sample. Because the sample is not random, it may contain biases or fail to generalize well to other populations. However, these data can complement other data sets or be used as benchmarks against them. – Data come mostly from large employers; medium and small firms may be underrepresented, although the MarketScan Research Databases include a large amount of data contributed from health plans."

RESPONSE: Thank you for the comment! Just as all other datasets, the MarketScan dataset is subject to a set of ascertainment criteria. Our conclusions are valid for the population the dataset represents. Yet, we need to proceed with caution when generalizing the results to other populations. For example, MarketScan only compiles insurance claim information from employer sponsored health insurance plan and does not include individuals on Medicaid or Medicare. This limitation has been previously described in an analysis based on MarketScan database by Wang et al. [6]. We had already mentioned this as part of the limitations for our study. Moreover, the heritability estimates we obtain is concordant with other phenome-wide studies (Table 2 and Results page 7 paragraph 3). In this revision, we further clarify it by adding the following sentences in the Discussion (Page 10 Paragraph 5).

“There are several aspects of our analyses that warrant discussion. First, MarketScan database only includes individuals with employer sponsored insurance policies. As such, low-income families may not be well-represented [6]. While the results are valid for the population the dataset represents, it is important to exercise caution when extrapolating our findings to different populations.”

5. Early onset disease versus late onset is also not a theme in the paper. It would be relevant to ask why this is not important in the context of the work presented.

RESPONSE: We agree that we have not addressed early-onset versus late-onset disease in the paper. As we define phenotypes based on PheWAS codes, we only can distinguish early-onset vs late-onset disease if they

each have their unique PheWAS codes. As mentioned earlier, we acknowledge that this is a limitation of all studies that rely on PheWAS codes.

Furthermore, as an insurance claim dataset, MarketScan only includes diagnostic information for a limited time. Thus, the prevalence of late-onset diseases is much lower in children when compared to parents. Therefore, to minimize the impact of this limitation, we conduct our analysis using the subset of nuclear families who were enrolled in the database for at least 6 years, and for whom all children are at least 10 years old at the time of entry into the database. For each phenotype, we also filtered out families where the age at enrollment of the youngest family member was less than the 5th percentile of the distribution of “age of diagnosis”. Lastly, we also included age and age² as covariates in both SMILE and SMILE-2 model. To clarify this, we added the following paragraph in Discussions about study limitations (Page 10 Paragraph 6):

“Second, as an insurance claim database, MarketScan data only includes medical information for a limited period of time [6]. For example, the data on children is only up to the age of 26, which is the maximum age a child can be covered by their parent’s health insurance in the US. Thus, the prevalence for late-onset conditions may be lower in children when compared to parents. We account for this by (1) limiting our analysis to families enrolled in the database for at least 6 years, (2) limiting our analysis to families where all children are at least 10 years old at the time of entry into the database, and (3) excluding families where the age at enrollment of the youngest family member was less than the 5th percentile of the age of diagnosis and (4) including age and age² as covariates in both SMILE and SMILE-2 mixed-effect linear models to account for the impact of age of onset. Despite the limitation of the datasets, our heritability estimates are comparable to estimates obtained from other data types (Table 2), which demonstrate the effectiveness of our filtering criteria and the validity of our results.”

6. I think the strength of the paper lies in the integration and handling of the exposure data, but I am not convinced that the statistical model, given the noisy definition of families and the variable validity of the diagnosis registration, generally can lead to a more accurate estimate of heritability compared to previous papers as the work states.

RESPONSE: We appreciate the comments above. We agree that the accuracy of genetic heritability estimations depend on the accuracy of inferred genetic relationships. Due to adoption or non-paternity, the inferred relationship may not be biological, which may bias the heritability estimates. In this revision, we conduct additional simulations to assess the robustness of heritability estimates if familial relationships in the data are mis-specified. We simulate 250,000 nuclear families, among which 2.4% have adoptive children and 6.2% have stepchildren, which reflects the estimates from census data. We evaluate the bias in the heritability. Our results showed that the mean squared error between true heritability and observed heritability estimates in the presence of adopted and stepchildren was only 0.00072 compared to 0.00035 without the presence of adopted or stepchildren. Given that the average length of 95% of confidence intervals is 0.032, the heritability estimates using inaccurate genetic relationships will still be covered by the confidence intervals of estimates using precise genetic relationships. The bias induced by relationship error is minimal. Our results are also concordant with the comparisons made in traditional family-based studies where they showed pedigrees with up to 20% errors led to only 5% underestimation of heritability [7, 8]. We have added a new sub-section called “*Impact of misaligned pedigrees and heritability estimates*” to the section of “*Robustness Analyses*” in the Supplementary Text (Page 13). We have also included Supplementary Figure 5 and Supplementary Table 3 to summarize the results.

Reviewer #2 (Remarks to the Author):

GENERAL COMMENTS

McGuire et al. conducted a simulation study examining how combining genetic, familial, and environmental variance components may help to better estimate genetic heritability of air pollution.

The study utilized data from several different resources, including electronic health records (EHR) was then linked with publicly available estimates of environmental data (including PM2.5 and NO2). A spatial mixed linear effect model (SMILE) was used to estimate heritability while incorporating both genetics and environmental variance components. The simulation study concluded that models including environmental variance parameters in the SMILE model provided better estimates of genetic heritability of air pollution. The study also conducted an instrumental variable analysis to estimate putative causal effect of PM2.5 and NO2 on various health outcomes. A couple suggested associations for variants located in the *FAT3* and *NLRP3* loci were identified.

The manuscript is overly complex and ambitious. However, the ambitious nature of the manuscript with the inclusion of several advanced methodologies detracts and hinders the manuscript's clarity, interpretation and overall motive. Several methodological decisions made were not clarified and/or substantiated. Additional justification for employing the methods described is severely needed (especially for the instrumental variable analysis). Additional comments are provided below.

We appreciate the thoughtful comments and suggestions. We have addressed all the comments below, which we believe strengthened our manuscript.

SPECIFIC COMMENTS

Introduction

The authors mention the issues with traditional genome wide association studies (GWAS) but fail to recognize the potential benefit of genome wide interaction studies (GWIS) in examining the interaction between both genetics and environmental factors on identifying associated loci.

RESPONSE: Thank you for the comment! In the revision, we clarified the differences between our approach and GWIS and the added value of our approach to improving heritability estimation. In brief, GWIS analysis uses explicitly measured environmental variables and studies how individual genetic variants interact with these environmental measures. In practice, it is impossible to measure all environmental risk factors explicitly, so GWIS cannot be used to quantify the contributions of unmeasured environmental risk factors to the trait variation. On the other hand, SMILE uses geolocations as a proxy for spatially correlated environmental risk factors. In theory, it can capture how spatially correlated environmental risk factors contribute to trait variation without having to measure each environmental risk factor individually. SMILE just needs genetic relationship matrix as input and does not need genetic data which also differs from GWIS approach. As a result, our approach is uniquely suitable for analyzing insurance claim data, which has no accompanied genetic data. We have clarified this in our revision (Page 10 Paragraph 2).

Introduction

The use of the EHR database could have severe selection bias issues which were never addressed or mentioned in the paper.

RESPONSE: Thank you for the comment! As all other studies, MarketScan dataset is also subject to ascertainment. The conclusions we draw from analyzing these datasets are valid for the population the datasets represent and it is important to exercise caution when generalizing the conclusions to other population. Moreover, the heritability estimates we obtain is comparable to other studies (Table 2 and Results page 7 paragraph 3). In the revision, we further clarify and expand it by adding the following sentences under the paragraph of study limitations in Discussions (Page 9 Paragraph 4).

“There are several aspects of our analyses that warrant discussion. First, MarketScan database only includes individuals with employer sponsored insurance policies. As such, low-income families may not be well-

represented [6]. While the results are valid for the population the dataset represents, it is important to exercise caution when extrapolating our findings to different populations.”

Results

The manuscript further fails to account for potential measurement error issues in using publicly available estimates for pollution measures and applying them to individuals. Furthermore, the it is possible that the instrumental variable used in the analysis themselves are measured with error.

RESPONSE: Thank you for the comment! We agree with the reviewer that pollution levels and wind are measured with errors, and it is important to evaluate how these measurement errors may influence causal effect estimate. In the revision, we performed simulations by adding noise to the measured wind and pollution levels. For each location, we randomly sample wind and pollution from 10 nearest neighbors and use the sampled values as noisy measurements. Using sampled wind and pollution measurements does not impose any distributional assumptions on the measures and hence ensure that the simulated noise is realistic. We assessed how the added noise impacted type-1 error, power, and mean squared error (MSE) of the estimated log-odds ratios in causal inference analyses of SMILE-2. We also compared the results with a standard two-stage regression using independent individuals (IND-FE). We observed similar levels of type-1 error, a 3.17% and 3.59% decrease in power, and 3.99% and 5.93% increase in MSE respectively when noisy wind and pollution levels are used. However, the 95% confidence intervals of power and MSE using noisy pollution and wind still overlapped with that using observed pollution and wind measurements. Importantly, SMILE-2 model continued to be more powerful and had lower MSE when compared to IND-FE models, supporting our main conclusion. Overall, this suggests that SMILE-2 is robust to measurement errors in pollution and wind. We have added a new sub-section called “*Impact of pollution and wind measurement errors on pollution causal effect estimates*” to the Supplementary Text (Page 13). We have also included Supplementary Figure 6 to summarize the results.

Statistical Methods Overview

It is unclear what the children-level family environment random effects represent, and how it would differ from parental level effects given nuclear families were used and I believe assume to have the same level of exposure.

RESPONSE: Thank you for the comment! We modeled children-level and parental-level environmental random effects separately since environments of parents and children can differ, including diet patterns, exercise levels, hours of sleep, and exposures at work or school. Thus, \mathbf{u}_{par} and \mathbf{u}_{child} are used to capture unique environment shared by parents and by children. This choice of random effects has been considered in other family-based association studies [6, 9]. In our revision, we have added the following sentences under “Statistical Methodology Overview” section in Results (Page 4 Paragraph 4).

“ \mathbf{u}_{par} and \mathbf{u}_{child} are the vectors of random effects for the shared parental and children-level family environment. Even though they live in the same household, parents and children may share distinct environment, including diet patterns, exercise levels, hours of sleep, and exposures at work or school, etc. \mathbf{u}_{par} and \mathbf{u}_{child} each capture the distinct environmental exposures that are shared between parents and between children.”

Statistical Methods Overview

The authors need to further clarify how the variance for the estimates in the model sampled with replacement was adjusted to account for the replacement of families.

RESPONSE: Thank you for the comment! The standard errors of our estimates are analytically derived using delta method. With the current sample size, performing bootstraps for all phenotypes is computationally prohibitive. We have provided detailed derivations in the Supplementary Text section “*Standard Errors for Liability-scale Variance Components by Multivariate Delta Method*” (Page 17). To further validate the accuracy

of the analytically derived standard error estimates, we compared our analytically derived standard error with bootstrap-based standard error for four randomly selected PheWAS codes with prevalence >10%. For each PheWAS code, we ran 100 bootstraps. We observed that the analytical and bootstrap-based standard error estimates were highly comparable (Table R1). This validates the accuracy of the analytically derived standard error estimates.

Table R1. Comparison of standard error derived analytically and empirically based on bootstrap.

PheWAS Code	Heritability Estimate	Heritability SE (analytical)	Heritability SE (empirical)
278	0.358	0.00397	0.00377
300.1	0.413	0.00387	0.00438
476	0.378	0.00359	0.00327
706.1	0.277	0.00523	0.00362

Estimating Causal Effects

The authors did not provide any assumptions that are made in an instrumental variable analysis and no evaluation regarding the instrumental variables and whether or not they meet the strict assumptions of an IV were conducted. The authors should present confidence intervals for their estimates.

RESPONSE: Thank you for the comment! We apologize for the confusion! We provided assumptions on the instrument variables in (Supplementary text page 21 paragraph 1). We have made it more clear in the revised manuscript by adding the assumptions to the method overview (page 5 paragraph 1). We had already provided two-sided p-values and odds ratio for SMILE-2 model pollution causal effects in Supplementary Table 2. As suggested, we have now also added the 95% confidence interval for pollution causal effects in Supplementary Table 2.

Reference:

1. Denny, J.C., et al., *PheWAS: demonstrating the feasibility of a phenome-wide scan to discover gene-disease associations*. *Bioinformatics*, 2010. **26**(9): p. 1205-10.
2. Denny, J.C., L. Bastarache, and D.M. Roden, *Phenome-Wide Association Studies as a Tool to Advance Precision Medicine*. *Annu Rev Genomics Hum Genet*, 2016. **17**: p. 353-73.
3. Ritchie, M.D., et al., *Robust replication of genotype-phenotype associations across multiple diseases in an electronic medical record*. *Am J Hum Genet*, 2010. **86**(4): p. 560-72.
4. Upadhyaya, S.G., et al., *Automated Diabetes Case Identification Using Electronic Health Record Data at a Tertiary Care Facility*. *Mayo Clin Proc Innov Qual Outcomes*, 2017. **1**(1): p. 100-110.
5. Saxena, R., et al., *Large-scale gene-centric meta-analysis across 39 studies identifies type 2 diabetes loci*. *Am J Hum Genet*, 2012. **90**(3): p. 410-25.
6. Wang, K., et al., *Classification of common human diseases derived from shared genetic and environmental determinants*. *Nat Genet*, 2017. **49**(9): p. 1319-1325.
7. Bérénos, C., et al., *Estimating quantitative genetic parameters in wild populations: a comparison of pedigree and genomic approaches*. *Mol Ecol*, 2014. **23**(14): p. 3434-51.
8. Charmantier, A. and D. Réale, *How do misassigned paternities affect the estimation of heritability in the wild?* *Mol Ecol*, 2005. **14**(9): p. 2839-50.
9. Xia, C., et al., *Pedigree- and SNP-Associated Genetics and Recent Environment are the Major Contributors to Anthropometric and Cardiometabolic Trait Variation*. *PLoS Genet*, 2016. **12**(2): p. e1005804.

REVIEWERS' COMMENTS

Reviewer #1 (Remarks to the Author):

I think the responses are generally fine and have strengthened the paper considerably. Especially the "Robustness Analyses" section and the extras in the SM. For the prescription based definition of T2D debut it is not really clear whether these are from primary care (GPs) or from secondary care medication data. If the latter it is less surprising that the results do not change much. At least a clarification of what kind of prescription data we are talking about would be useful. I may also have overlooked some detail around it. I think the authors have produced a nicely revised paper.

Reviewer #2 (Remarks to the Author):

Thank you for your thorough and complete responses to my initial concerns. I believe the edits made by the authors have greatly improved the clarity of the manuscript, and sufficient justification for the methods have been provided. I have no additional comments or concerns.

REVIEWER COMMENTS

Reviewer #1 (Remarks to the Author):

I think the responses are generally fine and have strengthened the paper considerably. Especially the “Robustness Analyses” section and the extras in the SM. For the prescription based definition of T2D debut it is not really clear whether these are from primary care (GPs) or from secondary care medication data. If the latter it is less surprising that the results do not change much. At least a clarification of what kind of prescription data we are talking about would be useful. I may also have overlooked some detail around it. I think the authors have produced a nicely revised paper.

RESPONSE: Thank you for the comment! We appreciate that the reviewer approves our responses to the initial revision and acknowledge that they have strengthened the paper considerably. We agree that whether the Type 2 Diabetes (T2D) prescription were given in a primary care or secondary care setting is an important aspect to consider when interpreting the results. Unfortunately, due to the limitations of the MarketScan electronic health record (EHR) data used, the information on the source of the prescriptions (primary vs. secondary care) is not available. We have now added a description in the sub-section called “*Impact of drug code on phenotype definition and heritability estimates*” to the section called “*Robustness Analyses*” in the Supplementary Methods (Page 11 paragraph 6) that clearly states this limitation: “*Due to the limitation of MarketScan EHR data, the source of the prescriptions (primary vs. secondary care) was not available and therefore was not used in our phenotype definitions*”. We believe this additional information would improve the transparency of our results and allow readers to understand the context of medication data used.

Reviewer #2 (Remarks to the Author):

Thank you for your thorough and complete responses to my initial concerns. I believe the edits made by the authors have greatly improved the clarity of the manuscript, and sufficient justification for the methods have been provided. I have no additional comments or concerns.

RESPONSE: Thank you for your thoughtful comments and suggestions that helped strengthen the paper and improve its clarity. We appreciate your time and effort.